# Proposal for the Design of Monitoring and Operating Irrigation Networks Based on IoT, Cloud Computing and Free Hardware Technologies

**DOI:** 10.3390/s19102318

**Published:** 2019-05-20

**Authors:** Luis Manuel Fernández-Ahumada, Jose Ramírez-Faz, Marcos Torres-Romero, Rafael López-Luque

**Affiliations:** 1Department of Computing and Numeric Analysis, University of Córdoba, Campus de Rabanales, 14071 Córdoba, Spain; lmfernandez@uco.es; 2Department of Electrical Engineering, University of Córdoba, Campus de Rabanales, 14071 Córdoba, Spain; marcos7396@gmail.com; 3Department of Applied Physics, University of Córdoba, Campus de Rabanales, 14071 Córdoba, Spain; fa1lolur@uco.es

**Keywords:** LPWAN, free hardware, sensor networks, IOT platform

## Abstract

In recent decades, considerable efforts have been devoted to process automation in agriculture. Regarding irrigation systems, this demand has found several difficulties, including the lack of communication networks and the large distances to electricity supply points. With the recent implementation of LPWAN wireless communication networks (SIGFOX, LoraWan, and NBIoT), and the expanding market of electronic controllers based on free software and hardware (i.e., Arduino, Raspberry, ESP, etc.) with low energy requirements, new perspectives have appeared for the automation of agricultural irrigation networks. This paper presents a low-cost solution for automatic cloud-based irrigation. In this paper, it is proposed the design of a node network based on microcontroller ESP32-Lora and Internet connection through SIGFOX network. The results obtained show the stability and robustness of the designed system.

## 1. Introduction and Background

In modern farms, the competitiveness of the sector and the growing demand for food [1] leads to an increase in water consumption, which requires optimal water management strategies ([2,3]).

The optimization of irrigation water consumption entails the improvement of crop development conditions through irrigation planning: water quantity and scheduling. For this purpose, automatic systems for monitoring variables and taking decisions are required. 

### 1.1. State of the Art

The need for optimization in agriculture became a reality in the 1970s. At the beginning, non-standardized wired electronic solutions designed ex professo were used. Since then, the development of irrigation system optimization has been closely linked to the rise and evolution of ICT [4].

There has been a trend in the last decade to implement intelligent irrigation management systems based on Wireless Sensor Networks (WSNs) [5], which have also been used in other agriculture areas. The characteristics and potentialities of WSNs perfectly match irrigation needs [6]. Presently, the integration of the devices to local communication networks, as well as to storing platforms, big data and information processing in the cloud, enable them to interact with other networks and the environment. [7,8] propose applications of IoT and Cloud Computing in agriculture

Among the advantages of WSNs, we can highlight the option of automating irrigation, since networks allow taking measurements (humidity, temperature, irradiance, etc.) and actions (solenoid valves, pumps) through the different independent electronic systems (nodes) that compose these networks. Several authors [5,9,10] have developed methodologies for the analysis and development of efficient networks based on needs assessment for different crops, soil attributes, climate, etc. In [11] the authors developed an application for smartphones to programme urban lawn irrigation by using evapotranspiration data from weather stations. This proposal achieved a saving of 48% of water in comparison to the previous irrigation system. In [12] the authors monitored golf courses and greenhouses by running an Android management application using WSN. This kind of tools allows instant decision making for any event and from any place. 

The integration of WSN and decision-making systems enables the development of irrigation plans based on energy savings, available water or the reduction of greenhouse gases. In [13] the authors developed an application whose aim is to save costs in pumping stations by finding balances between the flow (the result of valve opening levels) and the pressure at different points of the system. For this purpose, the authors implemented contrasted heuristic models ([14,15,16]). In [17], a system aimed at saving energy in irrigation facilities by making real-time decisions based on data acquisition was designed through WSN. A fuzzy control to optimize WSN energy consumption by establishing when to transmit the data collected [18]. In [19], the authors designed a multi-objective optimization algorithm to maximize benefits in every irrigation plot while reducing greenhouse gas emissions. With the same purpose, we can find works such as that of [20] which are oriented towards the search for pumping and monitoring systems that are totally disconnected from the electricity grid system. This is achieved by means of a real time model that synchronizes the photovoltaic energy available with the requirements for each sector.

Thanks to the automatic storage and the analysis of the measured data, in [21] the authors proposed the elaboration of databases from sensor measurements allowing the user of the application to model the operation of the irrigation system. The deployment of these solutions based on real-time data availability requires wireless communication networks both at local level and to the cloud.

Networks of distributed nodes in fields and farms require specific attention since the quality of transmissions among devices needs to be ensured. The vegetative state of the crop canopy or the shape of the soil are known to affect the information loss rates in radio wave transmissions. In this sense, the works developed in ([22,23]) must be highlighted.

### 1.2. Typology of Communication Networks 

Communications networks are classified into three different groups according to their range and transmission speed as shown in Figure 1 ([24,25]), namely:Short-range networks.Cellular networks.Long-range networks.

Short-range networks do not require a license for use. This group includes ZigBee or Bluetooth, among others. They are characterized by their low power consumption and high data exchange speed, although these features are limited to a short range (Table 1). In [21] the authors used this sort of networks for monitoring the irrigation infrastructure. 

Cellular networks outstand for a high transmission speed including short (Wi-Fi) and long-range with license (GSM, GPRS, 3G, 4G, 5G). In [26] a tracking system for sprayers in vineyards based on GSM and GPRS high data flow was developed.

Long-range networks present as advantages to the previous two systems that they have lower energy consumption, lower cost and a greater range.

A new model of long-range wireless networks, Low Power Wide Area Networks (LPWANs), has been recently developed. Its main feature is to have a star topology which has led to two models of technological development [28]. On the one hand, several telecommunications companies have used existing facilities (antennas, relay stations, etc.) to offer coverage to devices compatible with this technology [29]. On the other, collaborative networks have emerged for the global integration of IoT applications through low-cost open tools [30]. 

This “ease of use” increases since electronics companies, aware of the impulse of LPWANs, have incorporated compatible communication modules to the already known boards, thus facilitating connectivity (e.g., Arduino MKR 1200, Arduino MKR 1300, Pycom Lopy 4, etc.). 

LPWANs are very promising for the development of a monitoring, control and computing solution “in the cloud” in agriculture. Relevant variables are well known and, in most cases, have reaction times that limit the measurement frequency required (temperature, soil humidity, atmospheric humidity, precipitation, solar radiation, wind, hail). For this reason, the information transfer rate is low, and long-range communications are required [31].

The main aspects of SIGFOX, LoraWan and NBIoT are presented below considering their applicability in the digitalization of irrigation exploitations

*SIGFOX*: It is presented as the most limited option in terms of transmission speed (100 bps) and asymmetry, as it allows 140 uplink and 4 downlink messages per day. The uploading of messages admits up to 12 bytes for payload and up to 8 bytes for download. Moreover, the network use involves a cost per device similar to the use of the GPRS network in M2M (Machine to Machine) mode. Despite these limitations, SIGFOX is considered to be a suitable option for a wide range of agricultural projects since no additional facilities are required to be deployed and it achieves a high coverage in many EU countries (Figure 2). Several devices incorporating the SIGFOX communication module have emerged on the market due to the rapid growth of this technology.

*LoraWan*: this option has significant advantages in terms of operating costs, no costs associated with the use of radio space, and symmetry in communication [32]. While there is a 1%-time limitation on band occupancy, the transfer rate required in the applications analysed is not a constraint. The number of messages in both directions is a favourable aspect. The Things Network (TTN) platform that channels information from any LoraWan node has allowed an expansion of this type of network [30]. As an example, Figure 3 shows the current status of gateway deployment assigned to TTN in Spain.

Considering that the areas covered by the gateways are, on average, circles with a radius of 20 km, the availability of the network in agricultural areas is still very low. The growth in coverage is the result of ad hoc gateways implemented by users of the network.

*NBIOT:* This technology employs restricted frequencies which require a license for use. Its deployment is being carried out by the main telephone operators of each country (Vodafone, T-Mobile, AT&T, etc.). NBIOT provides a higher transmission speed than the other LPWAN options, although it involves a service cost per use and the range is reduced. Moreover, the availability of devices is still limited.

Figure 4 presents a qualitative comparison of the three standards analysed above.

Free hardware philosophy has been used in LPWAN network applications ([33,34]), and more specifically ATMEL microcontrollers compatible with its development environment (IDE) [35]. Its widespread popularity has allowed the development of boards adapted to Bluetooth communications, Wi-Fi, Ethernet, LoRa, LORAWAN, SIGFOX, or Android, as well as a catalogue of sensors to interact with and measure any variable. These features, together with its low cost and size, have led to its gradual integration in the field of WSNs ([36,37]).

### 1.3. IoT Platforms

The literature shows three possibilities to approach the implementation of monitoring and performance based on IoT and Cloud Computing. Firstly, specific programming models can be referred for specific problems, as the ones used for precision agriculture management [7] or for the control of irrigation valves [38]. This method entails a high programming effort, as it requires the coding of the entire application without reuse of code or adaptation to standard software. Secondly, we find commercial solutions adapted to the client’s needs in terms of data measuring and uploading to the cloud ([39,40]). As a drawback, these solutions are closed to the user. Finally, another way to address these technological challenges is to work with generic commercial tools (IoT platforms) where developers adapt the application to their specific needs. The report presents the most representative IoT platforms and their distribution in the market (Figure 5). 

This paper presents a new system for monitoring and acting on interesting variables in irrigation management. Economic competitiveness, standardization, and the use of flexible platforms are the key considerations for this design, which is based on the use of WSN and LPWAN communications, as well as on IoT platform integration. The solution presented here uses LoRa communications at the local level, and SIGFOX communications to access IoT platforms. At the hardware level, free hardware boards compatible with Arduino IDE have been used.

The structure of the article is as follows. Section 2 describes the choice of the control board, the communication system and the monitoring platform, as well as their development and programming. Section 3 details the implementation of the proposed system and the results obtained. Finally, Section 4 presents the conclusions of this work and the further lines of research to be developed.

## 2. Materials and Methods

### 2.1. Configuration of the Local System

#### 2.1.1. Local Network Architecture of the Agricultural Farm

When approaching the system architecture at a local level, we find two alternative topologies: 1-Direct access from each node to the cloud: Each node accesses the cloud directly through the available network (Figure 6a). It must be equipped with the communication module that supports the working network (SIGFOX, LORAWAN, etc.). This method does not include communication between nodes, but only vertical communication to the cloud - therefore, it is not necessary to develop any type if infrastructure. When opting for a network that requires pay-per-use in applications with a high number of nodes, this may represent a considerable operating cost.2-Access to a local concentrator: A star, tree or mesh local network links to the Internet through a local gateway equipped with a module compatible with the working network (Figure 6b). This solution is conditioned by the distance between nodes and data traffic requirements. The most suitable wireless communication technologies for local network are Wi-Fi, Bluetooth, Zigbee and Lora [9]. Table 1 shows the main characteristics of each technology.

The topology represented in Figure 6b has been considered for the solution presented here. As a result, only a single device connected to the cloud is required, thus reducing the cost of acquiring and operating modules. A solution is proposed based on sensor nodes distributed over the irrigation surface with a concentrator or gateway next to the pumping station, as shown in Figure 7.

For the selected topology, there are several communication alternatives available as described in Table 1. LoRa has been selected for this proposal as it provides much greater ranges than the other options. Thus, the result of communication consists of a LoRa network for the local level. Figure 7 shows the local nodes (hydrants and monitors) connected by LoRa technology to a gateway, consisting of a LoRa module. The connection to the cloud is detailed in Section 2.1.5. Since the longest distance between nodes in the working farm is 900 m (Figure 8), the proposed solution is adequate in according to features shown Table 1 (transmission range, [42]. 

#### 2.1.2. LoRa Hardware Platform for Local Network

Table 2 shows the characteristics for currently available electronic boards incorporating LoRa.

The HELTEC WiFi LoRa 32 board (Figure 9) is recommended for this work because its performance is adequate and the price is the lowest of all options—a relevant consideration when the number of measuring points is large. Moreover, this board incorporates a 0.96-inch OLED display, where messages established via software can be visualized.

The fieldwork design of the LoRa network for this paper has been developed by experimentally testing the signal level in both the receiver and the transmitter (no package loss). It has not been considered the potential interference between the foliar mass growth of the olive tree and the quality parameters of the radio information transmission [22] since the olive tree is a low-density canopy species. For all nodes 1.2 m height antennas are available (3m height for gateway antenna). Olive trees in orchard frames are isolated (Figure 8 and Figure 10). This feature together with the low leaf density enables successful communications between nodes. The antenna used is stubby type, quad band. Its features are: Length: 58 mm; Impedance: 50 Ω; Gain: 2 dBi (3.5 dBi peak); VSWR: 2 maximum; Polarisation: Vertical; Connector: SMA Male; Directivity: Omnidirectional.

#### 2.1.3. Sensors

There are sensors to measure the whole range of relevant irrigation variables: hydraulic network (pressures and flows), environmental variables (i.e., air temperature and humidity, irradiance, wind speed, precipitation), and soil variables (i.e., humidity, temperature, pH, matric potential). 

In this work, irrigation management has been approached exclusively from soil moisture, but there are other criteria to control irrigation. In this case, a sensor is required to provide soil moisture, and its supply voltage and measurement signal must be compatible with HELTEC Wi-Fi LoRa 32 board. Under these conditions, SHT 15 (manufactured by Sensirion) sensor has been chosen. It is encapsulated in a sintered metal enclosure protecting electronics from direct contact with water (see Figure 11).

The power supply, 3.3 V, can be obtained from the electronic board. Sensor consumption is 0.9 mA. The accuracy offered is +−2% in the range of 10–90 % relative humidity, and +−4% out of this range.

#### 2.1.4. Power Supply for HELTEC Wi-Fi LoRa 32 Board

Power supply is an important issue in the agriculture, since its availability is not frequent in locations to be sensed. 

It is essential to know the energy demands to address the problem of power supply. This requirement considers three states of the board: running, transmitting and deep sleep. In running mode, the microprocessor reads the sensor value. In transmitting mode, data are sent to the concentrator or gateway. Once this process has been completed, it switches to a low-power mode until the next measurement is taken. This mode involves the deactivation, via software, of all the elements comprising the microcontroller, with the exception of the main timer. In this mode, the CPU and the LoRa and Wi-Fi radio modules are disconnected.

The board was monitored by executing the work cycle above using a YOKOGAWA DL850E oscilloscope, obtaining the results shown in Figure 12. The most significant values are I_r_, I_ta_, I_dp_ and t_t_ (shown in Table 3), where: I_r_ = Running mode current (mA)I_ta_ = Transmitting mode average current (mA)I_ds_ = Deep sleep mode current (mA)t_t_ = Transmitting time (s)

The time that the system remains in deep sleep mode is given by the frequency of the measurements to be made.

A second delay is applied to stabilize sensor measurements when the deep sleep mode ends. Table 4 shows, for different measurement frequencies, the required load values per working day.

From data shown in Table 4, the influence of daily consumption measurement frequency is practically insignificant, with the highest consumption during deep sleep periods. If power is supplied by means of lithium ion cells type 18,650, with a unit capacity of 3.000 mAh, in the case of using a single cell, the charge duration would be just over ten days. Even with several cells in parallel, duration would not be enough for an autonomous operation application. Based on these data, two alternatives are considered: (1) energy generation system based on a photovoltaic cell for battery recharging, (2) replacement of deep sleep mode by disconnection, via hardware, of the entire equipment. 

The first option, based on energy generation by photovoltaic cell, will require one or more 1 W-peak power photovoltaic modules with a 6 V open-circuit voltage of and 200 mA short-circuit current. To design the number of modules and cells, the method proposed by Posadillo and López Luque [43] using the LLP (Loss of Load Probability) concept is applied. A 0° angle module inclination has been assumed as the most disadvantageous option. The results obtained are shown in Table 5.

The most suitable option (with the lowest probability of failure) is that with two cells in parallel and one module due to set compactness.

The TPL5110 module, a low-power timer, is proposed for the second option, based on replacing deep sleep mode with hardware disconnection, which interrupts power supply for a period of time from 100 ms to 7200 s. During interruption time, consumption is reduced to 25 µA. Table 6 shows consumption under different scenarios.

According to Table 6, when supplying the system with two lithium ion cells with a nominal capacity of 3000 mAh, the battery life would be over 6 years in the most unfavorable case (15-min sending frequency). 

#### 2.1.5. Local Gateway

Table 7 specifies the technological characteristics for SIGFOX boards (microcontroller and module). These boards are flexible and have numerous analogue and digital inputs and outputs.

The Arduino MKR1200 module has been selected in this work because it is programmed in a well-known environment (Arduino IDE) and used by a wide community of developers that freely share libraries and resources for programming. Power supply for this board is not considered here because it is located in the pumping station of the farm, where electrical network is available. A wired serial communication is set between the Heltec ESP32 LoRa board, operating as a hub for the set of nodes distributed in the farm, and the module MKR1200, which acts as a local gateway. The module MKR1200 communicates to the cloud through SIGFOX.

### 2.2. Exchanging Information with the Cloud

Concerning the communications required for irrigation management, the main aspects are: coverage, scope, cost and energy consumption of electronics. Figure 4 shows how SIGFOX improves LORAWAN and NBIoT features. 

SIGFOX uses Ultra Narrow Band communications encoded by DBPSK (Differential Binary Phase Shift-Keying) and optimized for the exchange of short messages. The protocol accepts messages up to 38 bytes for a 12-byte payload. Figure 13 shows the upload message structure.

The limitation of the payload to 12 bytes for uplink messages and 8 bytes for downlink messages raises the need to design a strategy for sending and receiving information. The number of daily upload messages is limited to 140, and the number of daily download messages is limited to 4. Figure 14 shows the download message structure.

Soil moisture is a value varying between 0 and 100, so it can be represented by a 7-bit binary number (0…127). Thus, it is possible to include 13 values in each transmission. The remaining 5 bits are used to identify the group of nodes corresponding to the frame. If the number of nodes of the system is less than 14, humidity values can be updated every 15 min. For a higher number of nodes, there are two possibilities, either to update with a frequency of 30 min (14 to 27 nodes) or higher, or to have more than one local gateway.

The four download messages are used to establish an irrigation priority between the nodes and the application time. 

The operating time can vary from 0 to 2048. This time applies equally to all nodes, acting on those selected valves by assigning “1” to the corresponding bit.

The information sent by the MKR1200 module reaches the backend, a SIGFOX web application where users can visualize the data in a hexadecimal format list. The information must be sent in real time from the backend to a platform with a user interface for data consultation and processing, as well as for implementing decision-making processes when necessary. The method of transmitting information from the SIGFOX backend is called callback. Figure 15 shows SIGFOX communication architecture.

The callback can be routed to its own server or to an IoT platform. Thingspeak, an IoT platform, has been chosen in this paper considering its low infrastructure requirements. Data can be graphically displayed in the web application or in any mobile device [44]. In addition to recording and displaying information, this platform provides the mathematical analysis tool MATLAB^®^, enabling the user to perform all kinds of information processing, including generating actions based on the results, implementing analysis and decision-making programs.

## 3. Results

The main result of this research is the design and implementation of a comprehensive monitoring and a high performance system based on communication with the cloud for irrigation systems. Figure 16 shows the architecture of the solution.

The elements that compose the system are:Monitor node: This is a device consisting of a Heltec ESP32 LoRa board measuring soil moisture through a SHT15 sensor. As shown in Figure 17, the board is powered from the TLP5110 module and the battery pack. To ensure the durability of the electronic devices, the unit has been placed in a PVC box with IP67 protection. The node measures soil moisture every 15 min and sends the data via LoRa network to the local gateway. Once the operation has been performed, a digital output is activated to order the TPL5110 module to disconnect the power supply until the next transmission.Hydrant node: Designed similarly to the previous node at a hardware level, although it is not equipped with a humidity sensor. To act on the hydrant valve, digital output from the Heltec ESP32 LoRa board is used. It connects to the gateway every 15 min to request priority and time settings.Gateway or local concentrator: It consists of two devices, a Heltec ESP32 LoRa board and an MKR1200 board. The first one operates as a LoRa communications concentrator. Through this element communication is directed to hydrant nodes and from monitor nodes. This board communicates through the serial port with the MKR1200 board, which is the platform that communicates to the cloud.SIGFOX backend: The information sent by the local gateway reaches the SIGFOX web platform where data are sent to the IoT platform. This process is bidirectional, the backend receives data from the IoT platform to be sent to the MKR1200 board. Figure 18 shows data in the backend.IoT platform: ThingSpeak receives information from field sensors and records it in its own database. At the same time, an application developed in MATLAB^®^ analyses moisture data, and generates priority and run time settings for each hydrant node.

A practical application of the system proposed here has been implemented in an olive orchard located in Córdoba (Spain). Figure 19 shows the location of the farm. It is a 37.66 ha of olive tree farm (Olea europea, L.) in an 8 m × 8 m frame. The farm has drip irrigation (1 dripper/1.5 m). The trees are adult (45 years old). A 56,500 m^3^ reservoir, whose supply comes from a connection to the community of irrigators Genil-Cabra, pressurizes the farm by means of a 50 HP pumping station. Figure 8 shows how water is distributed through a branched network of polyethylene pipes to 3 sectors.

In October 2018, monitoring and control systems for three irrigation sectors were installed. Figure 8 also shows the layout of hydrant nodes, monitor and gateway. Figure 9 shows the layout of a monitor node in the field, while Figure 20 shows stored humidity data on the ThingSpeak platform. No transmission faults were detected during the test period.

Table 8 shows the cost of the total equipment.

Moreover, the annual operating cost of SIGFOX network is 16.53 €/gateway. The use of the ThingSpeak platform implies a cost of 600 €/year (Standard license) for a limit of 33 M data. Since a monitor node downloads 35,040 data/year, the Standard license would support a maximum of 900 nodes.

## 4. Conclusions

An IoT solution has been proved to be a solution to control and manage irrigation. This is a geographically scalable solution applicable to other areas of agriculture. Problems related to connectivity and energy availability have usually made automation difficult in rural areas. 

The emergence of SIGFOX technology with its low implementation and operating costs allows isolated devices to be connected to the cloud.

Furthermore, a strategy focused on reducing electronics energy consumption has enabled the nodes to have a power independence for over 5 years. 

ICT based on free hardware and software have led to a high reliability and low-cost solution either in terms of investment or operation. The solution has been tested in real conditions for four months. Robustness and stability of the communications and hardware with been verified. 

The IoT platform used in this work (ThingSpeak) has allowed interaction with controlled facilities from any device connected to the Internet.

The prolonged operation of systems such as the proposed here will generate a large volume of data from which knowledge will be extracted, as well as new irrigation strategies can also be developed.

## Figures and Tables

**Figure 1 sensors-19-02318-f001:**
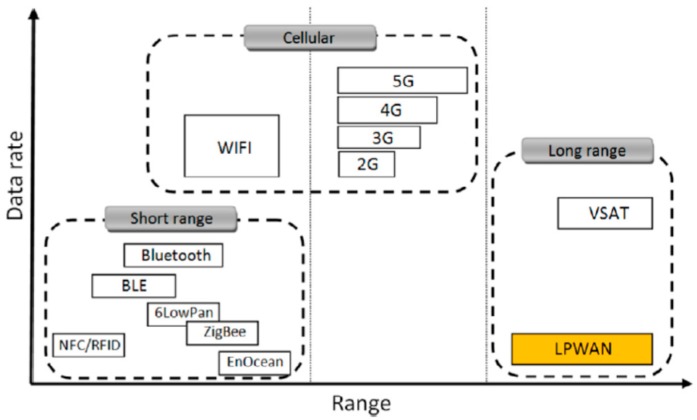
Data rate vs. range in communication networks. Source: [27].

**Figure 2 sensors-19-02318-f002:**
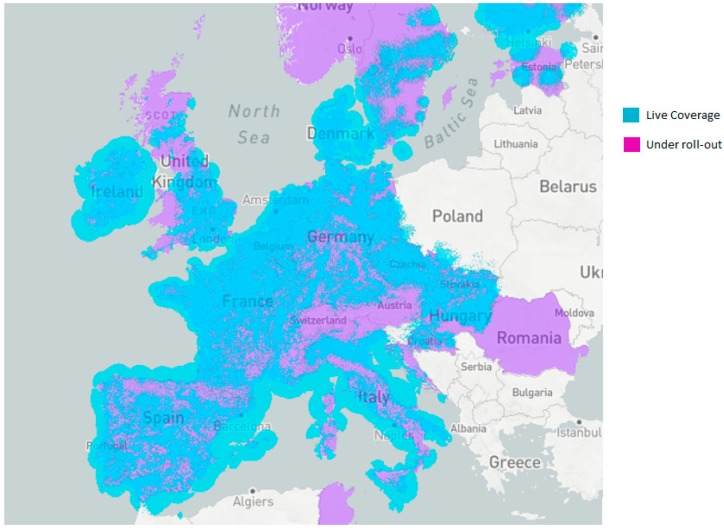
SIGFOX coverage. Reference: [29].

**Figure 3 sensors-19-02318-f003:**
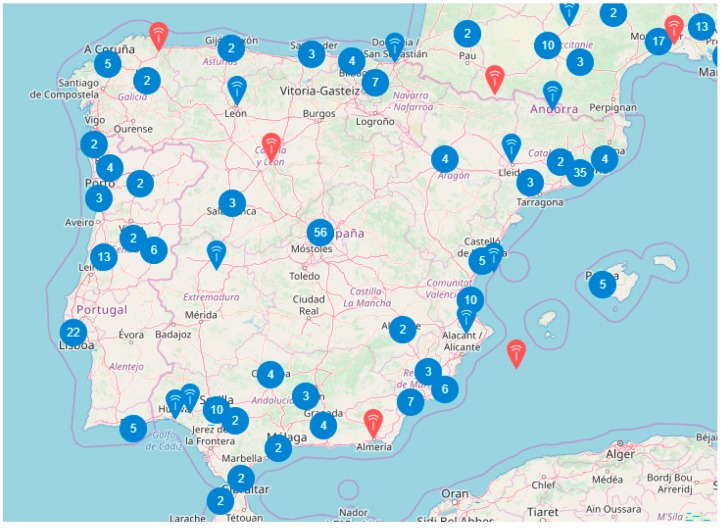
Gateways assigned to TTN in Spain. Source: [30].

**Figure 4 sensors-19-02318-f004:**
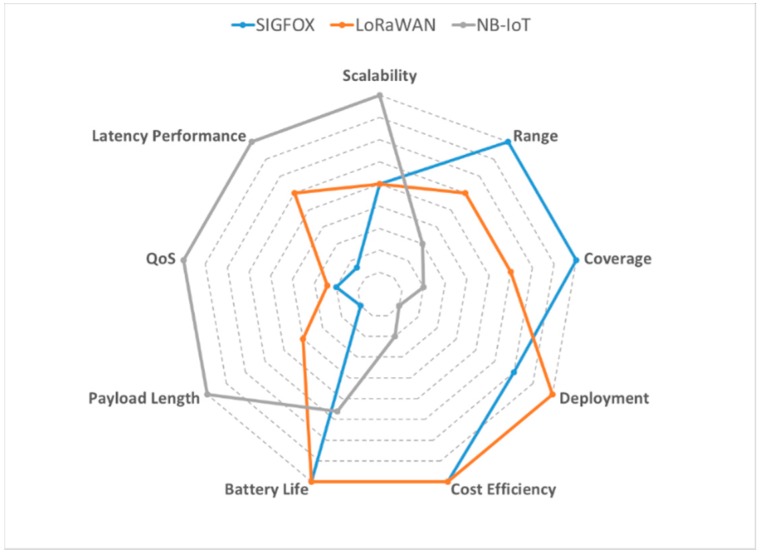
Qualitative comparison between SIGFOX, LORAWAN and NBIoT (Source: Own elaboration).

**Figure 5 sensors-19-02318-f005:**
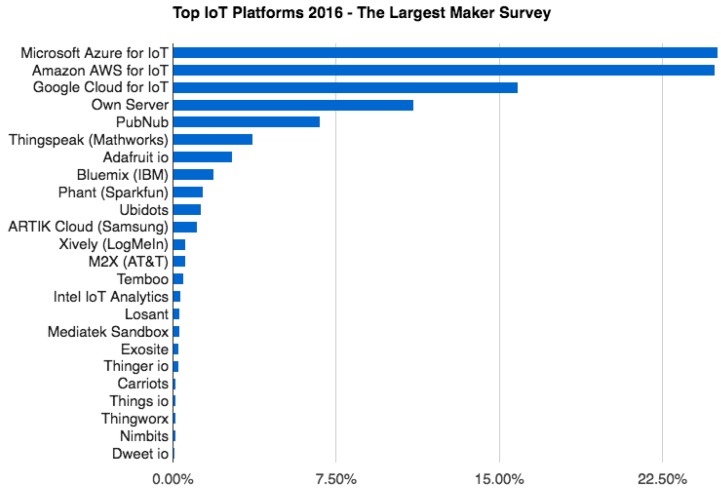
Top IoT platforms. Source: [41].

**Figure 6 sensors-19-02318-f006:**
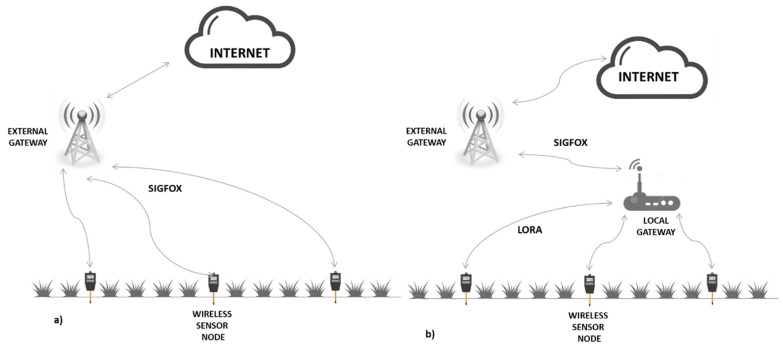
Local architecture configurations. Source: Own elaboration.

**Figure 7 sensors-19-02318-f007:**
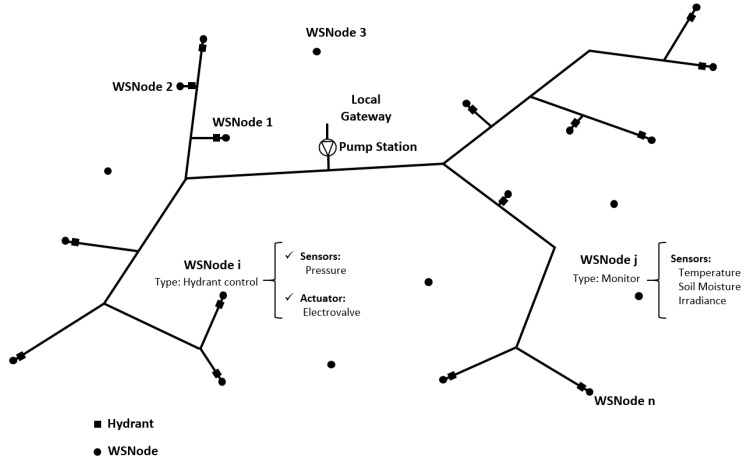
Proposal for node distribution and local gateway distribution. Source: Own elaboration.

**Figure 8 sensors-19-02318-f008:**
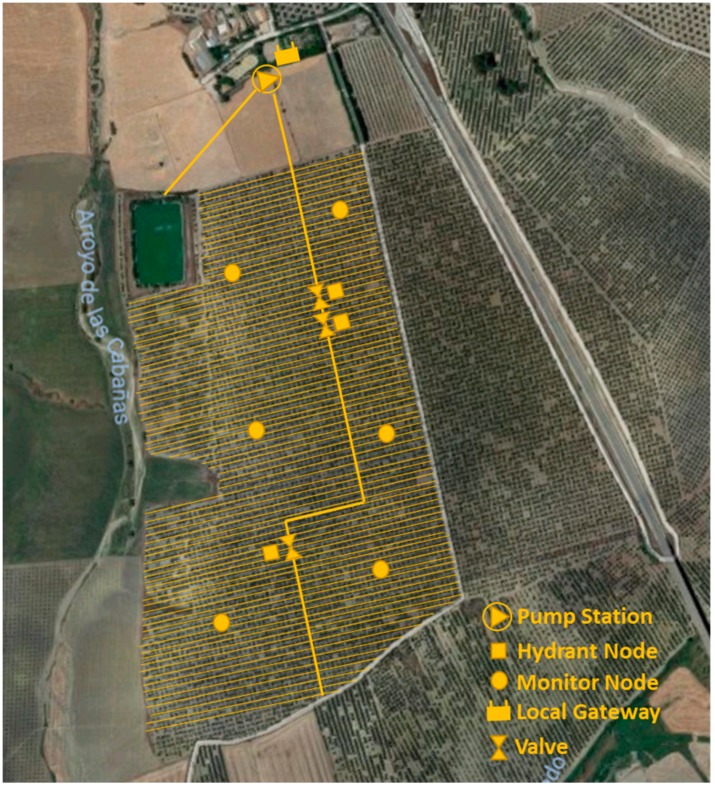
Farm irrigation and node distribution layout.

**Figure 9 sensors-19-02318-f009:**
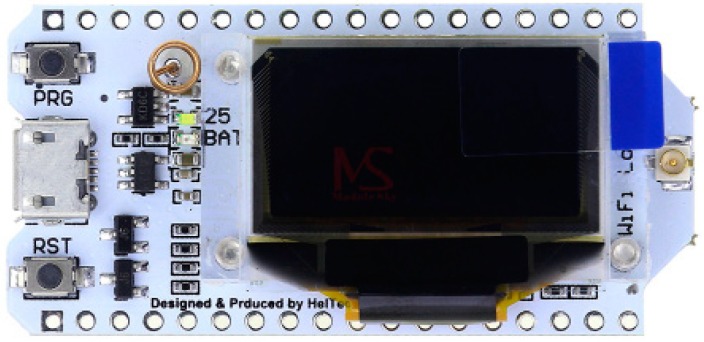
Heltec WiFi LoRa 32.

**Figure 10 sensors-19-02318-f010:**
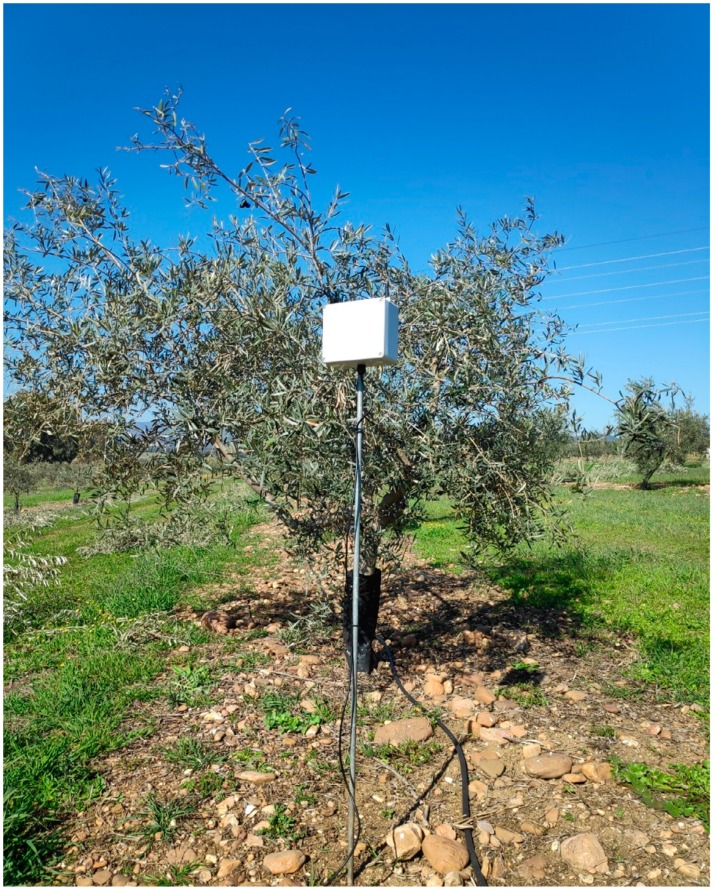
Field monitor node.

**Figure 11 sensors-19-02318-f011:**
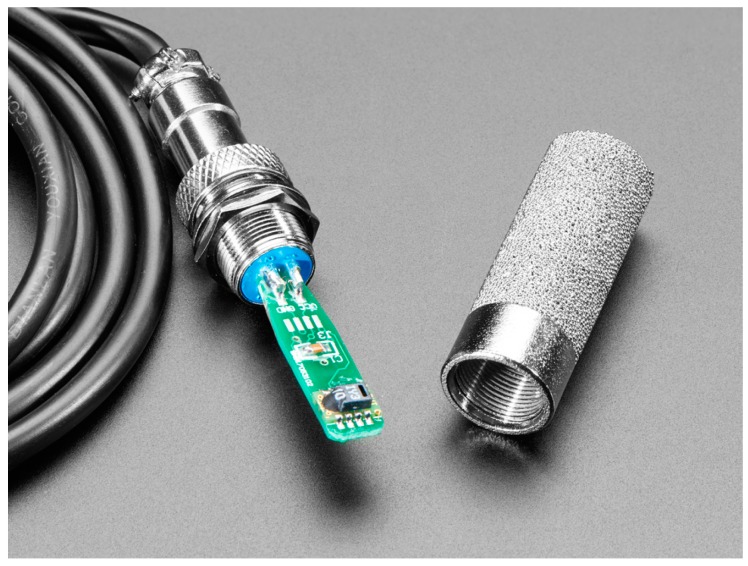
SHT15 sensor.

**Figure 12 sensors-19-02318-f012:**
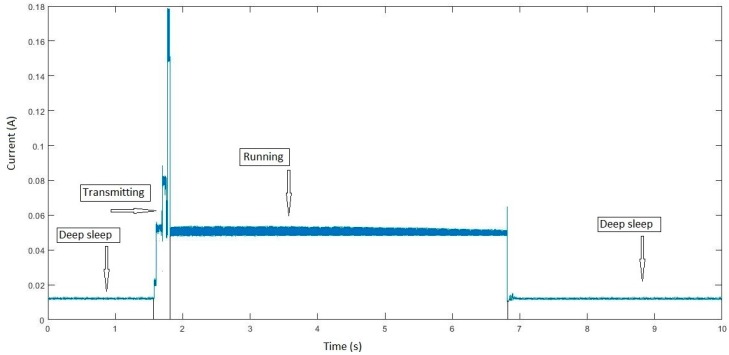
Electric current consumed by HELTEC Wi-Fi LoRa 32 in its three function modes.

**Figure 13 sensors-19-02318-f013:**
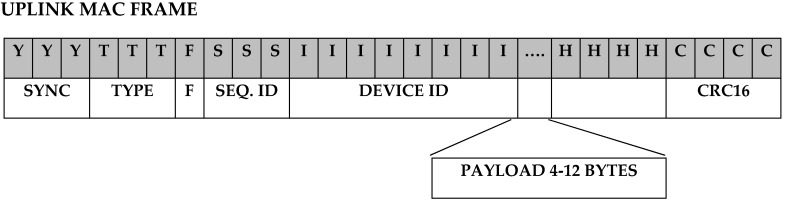
Upload message structure in SIGFOX.

**Figure 14 sensors-19-02318-f014:**
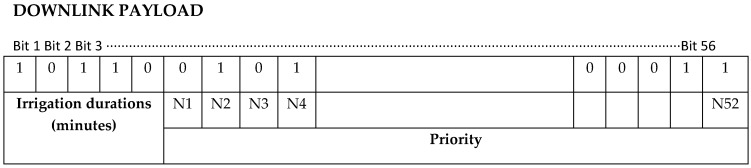
Download message structure in SIGFOX.

**Figure 15 sensors-19-02318-f015:**
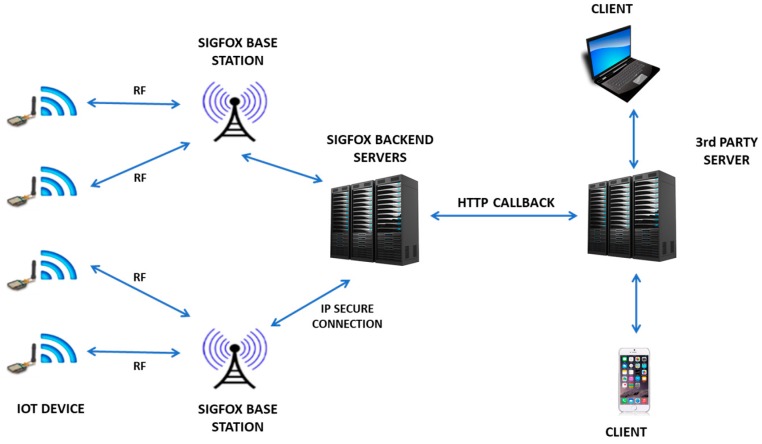
SIGFOX network topology.

**Figure 16 sensors-19-02318-f016:**
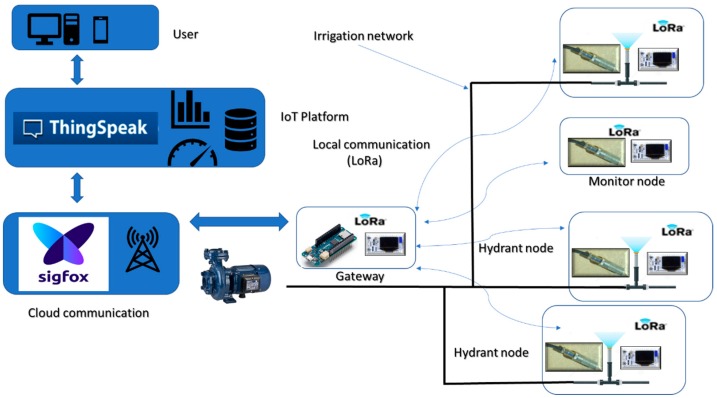
Global system architecture.

**Figure 17 sensors-19-02318-f017:**
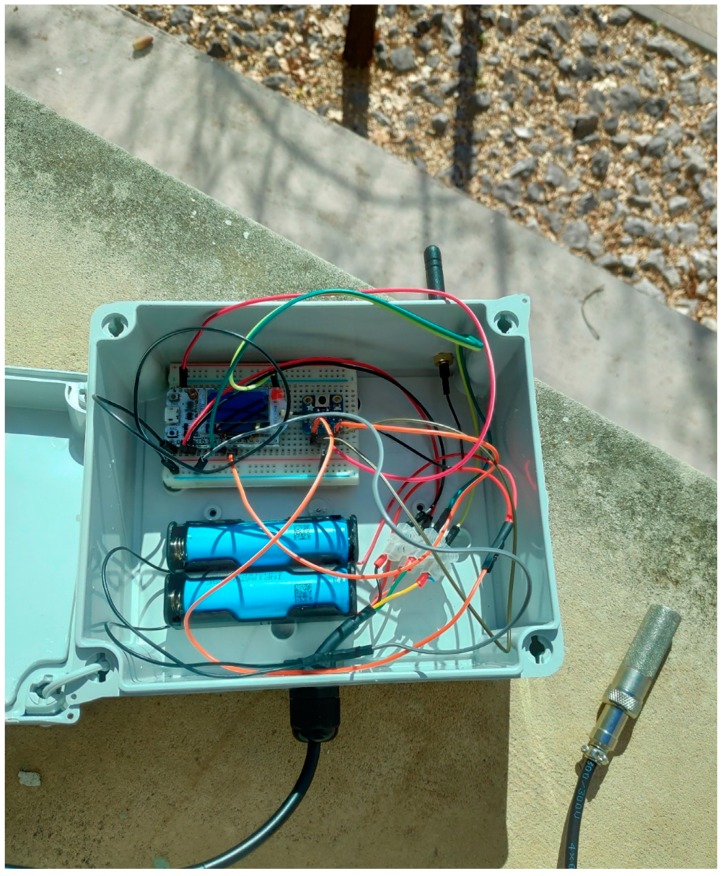
Monitor Node.

**Figure 18 sensors-19-02318-f018:**
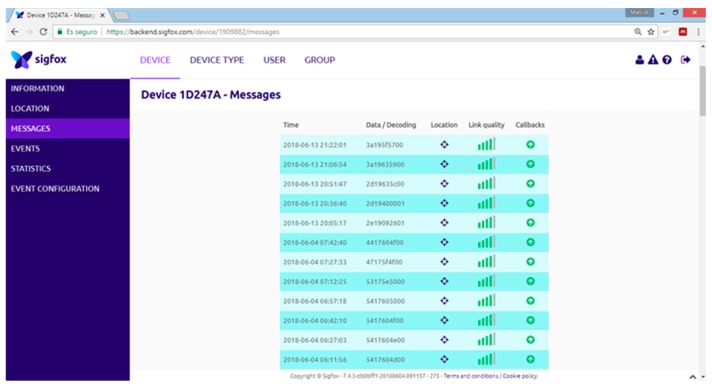
Messages sent in SIGFOX backend. Source: [29].

**Figure 19 sensors-19-02318-f019:**
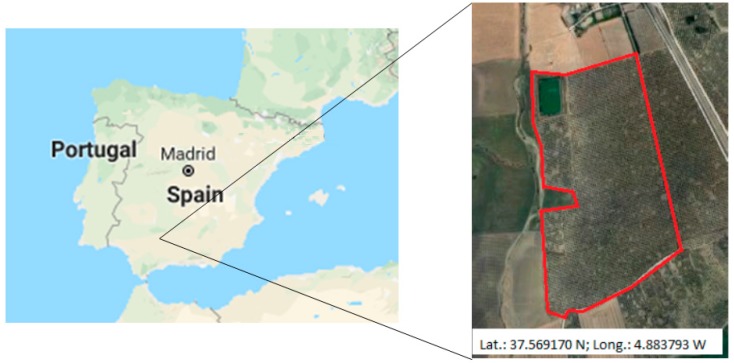
Farm location.

**Figure 20 sensors-19-02318-f020:**
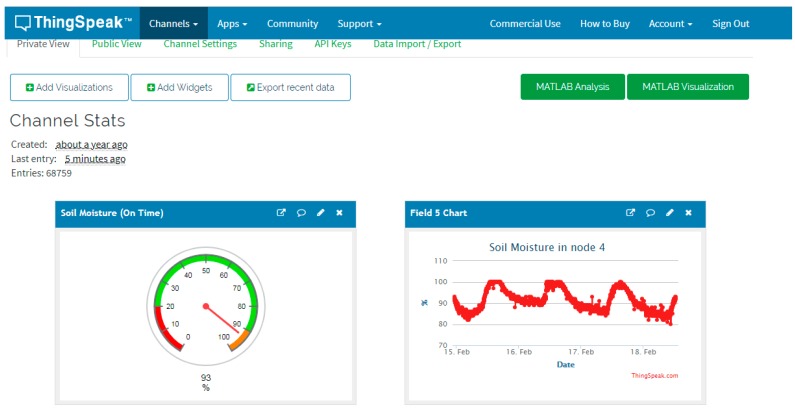
Monitored data at Thingspeak application.

**Table 1 sensors-19-02318-t001:** Local network communication technology characteristics.

Parameter	Wi-Fi	Bluetooth	Zigbee	Lora
Standard	IEEE 802.11 a,b,g,n	802.15.1	802.15.4	802.15.4g
Frequency	2,4 GHz	2,4 GHz	868/915 MHz, 2,4 GHz	433/868/915 MHz
Data rate	2–54 Mbps	1–24 Mbps	20–250 kbps	0.3–50 kbps
Transmission Range	20–100 m	8–10 m	10–20 m	>500 m [42]
Topology	Star	Star	Tree, star, mesh	Star
Energy consumption	High	Medium	Low	Very Low
Cost	Low	Low	Low	Low

**Table 2 sensors-19-02318-t002:** Technical characteristics of LoRa integrated boards.

	LoRa 32u4 II	Heltec WiFi LoRa 32 V2	SparkX SAMD21 Pro RF 1W LoRa	Pycom Lopy 4	TTGO LoRa32 T-Beam	TTGO LoRa32 V2.1
Microcontroller	ATMEGA 32u4	ESP32	SAMD21	ESP32	ESP32	ESP32
Programming	Arduino IDE Compatibility	Arduino IDE Compatibility	Arduino IDE Compatibility	MicroPython	Arduino IDE Compatibility	Arduino IDE Compatibility
Lora Chipset	SEMTECH SX1276	SEMTECH SX1276	SEMTECH SX1276	SEMTECH SX1276	SEMTECH SX1276	SEMTECH SX1276
Transmitting power	+20 dB	+20 dB	+30 dB	+20 dB	+20 dB	+20 dB
Operating frequency	868–915 MHz	868–915 MHz	868–915 MHz	868–915 MHz	868–915 MHz	868–915 MHz
ROM	32 kB	448 kB	256 kB	448 kB	448 kB	448 kB
RAM	2 kB	520 kB	32 kB	520 kB	520 kB	520 kB
Logic level	3.3 V	3.3 V	3.3 V	3.3 V	3.3 V	3.3 V
Analog input	10	18	5	18	18	10
Digital I/O	20	28	2	23	16	17
Transmit current	128 mA	146 mA		108 mA		
Standby current	11 mA	46 mA		35 mA		
Deep sleep current	300 µA	2,4 mA		1 µA		
Other Features	Wi-Fi, BLE	Wi-Fi, BLE, OLED display		Wi-Fi, BLE, SIGFOX	Wi-Fi, BLE, GPS, CANBus, SMA connector	OLED display, Wi-Fi, SMA connector
Price	30 €	12 €	45 €	35 €	20 €	20 €

**Table 3 sensors-19-02318-t003:** Electric parameters for running, transmitting and deep sleep modes.

I_r_ (mA)	I_ta_ (mA)	I_ds_ (mA)	t_t_ (s)
50.1	71.7	11.9	0.2384

**Table 4 sensors-19-02318-t004:** Daily energy consumption according to measurement frequency for each function mode.

Frequency (h)	Running Mode Consumption (mAh)	Transmitting Mode Consumption (mAh)	Deep Sleep Mode Consumption (mAh)	Total (mAh/day)
0.25	1.34	0.46	285.60	**287.39**
0.5	0.67	0.23	285.60	**286.50**
1	0.33	0.11	285.60	**286.05**
2	0.17	0.06	285.60	**285.82**
6	0.06	0.02	285.60	**285.67**
12	0.03	0.01	285.60	**285.64**
24	0.01	0.005	285.60	**285.62**

**Table 5 sensors-19-02318-t005:** LLP values for different configurations.

Number of Modules	Number of Parallel Cells	LLP (%)
1	1	0.3
1	2	0
2	1	0

**Table 6 sensors-19-02318-t006:** Daily energy consumption according to measurement frequency for each function mode.

Frequency (h)	Running Mode Consumption (mAh)	Transmitting Mode Consumption (mAh)	Disconnection Mode Consumption (mAh)	Total (mAh/day)
0.25	1.34	0.46	0.72	**2.51**
0.5	0.67	0.23	0.72	**1.62**
1	0.33	0.11	0.72	**1.17**
2	0.17	0.06	0.72	**0.94**
6	0.06	0.02	0.72	**0.79**
12	0.03	0.01	0.72	**0.76**
24	0.01	0.005	0.72	**0.74**

**Table 7 sensors-19-02318-t007:** Technical characteristics for SIGFOX integrated boards.

	MTDuino-SFM2CWW001	Pycom SiPy	Arduino MKR 1200
Microcontroller	ATSAMD21	ESP32	ATSAMD21
Programming	Arduino IDE Compatibility	MicroPython	Arduino IDE Compatibility
Transmitting power		14/22 dBm (Europe/America)	
Operating frequency	868–915 MHz	868–915 MHz	868 MHz
ROM	256 kB	4 MB	256 kB
RAM	32 kB	512 kB	32 kB
Logic level	3.3 V	2.2 V	3.3 V
Analog input	6	8	7
Digital I/O	20	24	20
Other Features		Wi-Fi, BLE	

**Table 8 sensors-19-02318-t008:** Total equipment costs.

Concept	Node Type Monitor	Node Type Hydrant	Gateway
Placa Heltec Wi-Fi LoRa 32 V2	€20	€20	€20
MKR1200 board	-	-	€35
SHT15 Soil Moisture sensor	€20	-	-
TPL5110 timer	€4	€4	-
Li-Ion Battery	€5	€5	-
Relay module	-	€2	-
Enclosure	€8	€8	€12
AC DC adapter	-	-	€3
Others	€2	€2	€2
Total costs	€59	€41	€72

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
