# Peer review of "Proposal for the Design of Monitoring and Operating Irrigation Networks Based on IoT, Cloud Computing and Free Hardware Technologies"

_sensors, 2019, doi:10.3390/s19102318_

Round 1

Reviewer 1 Report

A radioelectric study (even short or simplified) should be accomplished in order to determine:

Maximum and minimum range between nodes.

Terrain influence. The network works under the conditions of a near-ground scenario which has been demonstrated to influence the packet loss, maximum data rate, BER, and range between nodes.

Vegetation influence (especially during the blooming season).

Justify the reason supporting the chosen antenna heights (avoiding near-ground, 2-ray multipath) providing brief maths.

Indicate antenna features, transmitted power, and other parameters needed for potential readers.

Complete the references according to the required improvements. Same apply to the discussion of results.

Author Response

RESPONSE TO REVIEWER 1 COMMENTS (SENSORS 500616)

·         Comment 1: “Maximum and minimum range between nodes”.

We have added the complete reference missing from Table 1 (Ausgustin et al., 2016) and referring to the distance range at which LoRa technology can be used. We have also presented the maximum distances used in the application described in the article. The complete sentence would be (Section 2.1.1): “Since the longest distance between nodes in the working farm is 900 meters (Figure 18), the proposed solution is adequate in according to features shown Table 1 (transmission range, (Augustin et al., 2016). .

·         Comments 2 and 3: “Terrain influence. The network works under the conditions of a near-ground scenario which has been demonstrated to influence the packet loss, maximum data rate, BER, and range between nodes”; “Vegetation influence (especially during the blooming season)”.

In this article, the results obtained empirically show a very high level of success in communications. Both in the response to this comment and in the following one, we refer to the works dealing with this problem, although they are not applicable to our study in some cases due to the specific restrictions of olive groves. The sentence added is (Section 1.1): “Networks of distributed nodes in fields and farms require specific attention since the quality of transmissions among devices needs to be ensured. The vegetative state of the crop canopy or the shape of the soil are known to affect the information loss rates in radio wave transmissions. In this sense, the works developed by Jebril et al. (2018) and Iova et al. (2017) must be highlighted.”.

·         Comment 4: “Justify the reason supporting the chosen antenna heights (avoiding near-ground, 2-ray multipath) providing brief maths”.

A holistic consideration of the work installation features has been realized in order to contextualise the added comments on the use of antennas. The whole sentence would be (in Section 2.1.2): The fieldwork design of the LoRa network for this paper has been developed by experimentally testing the signal level in both the receiver and the transmitter (no package loss). It has not been considered the potential interference between the foliar mass growth of the olive tree and the quality parameters of the radio information transmission (Jebril et al., 2018) since the olive tree is a low-density canopy species. For all nodes 1.2m height antennas are available (3m height for gateway antenna). Olive trees in orchard frames are isolated (Figures 18 and 19). This feature together with the low leaf density enables successful communications between nodes.”

·         Comment 5: “Indicate antenna features, transmitted power, and other parameters needed for potential readers”.

In Section 2.1.2 these features are added:

The antenna used is stubby type, quad band. Its features are:

ü  Length: 58 mm

ü  Impedance: 50 Ω 

ü  Gain: 2 dBi (3.5 dBi peak)

ü  VSWR: 2 maximum

ü  Polarisation: Vertical 

ü   Connector: SMA Male

ü  Directivity: Omnidirectional

·         Comment 6: “Complete the references according to the required improvements. Same apply to the discussion of results.”

Three new references have been added:

ü  Augustin, A., Yi, J., Clausen, T., Townsley, W., 2016. A Study of LoRa: Long Range & Low Power Networks for the Internet of Things. Sensors 16, 1466. doi:10.3390/s16091466

ü  Iova, O., Murphy, A., Picco, G. Pietro, Ghiro, L., Molteni, D., Ossi, F., Cagnacci, F., 2017. LoRa from the City to the Mountains : Exploration of Hardware and Environmental Factors, in: Proceedings of the 2017 International Conference on Embedded Wireless Systems and Networks. pp. 317–322.

ü  Jebril, A.H., Sali, A., Ismail, A., Rasid, M.F.A., 2018. Overcoming limitations of LoRa physical layer in image transmission. Sensors (Switzerland) 18. doi:10.3390/s18103257

Reviewer 2 Report

Very interesting work about the application of LPWAN technology
in automated irrigation systems.

Some comments must be taken into account:

In the abstract improve the syntax of the phrase "This work
presents a very low-cost solution for irrigation automation based on the cloud.."

Define the color usage in Fig. 2.

Explain in more detail the usage of both LoRA and SIGFOX (Local star topology
for LoRA and Sigfox for the LoRA GATEWAY?) The solution presented here uses
LoRa communications at the local level, and SIGFOX communications to access IoT platforms..

Correct in Fig. 6 a number "187"

correct the phrase "of a comprehensive monitoring and performance system..."
you mean "high performance"?

maybe the phrase "The IoT platform used in this work (ThingSpeak) integrates the MATLAB® tool which allows
the implementation of analysis and decision-making programs.."
must be removed from the conclusions section to other section of the article.

Author Response

RESPONSE TO REVIEWER 2 COMMENTS (SENSORS 500616)

·         Comment 1: “In the abstract improve the syntax of the phrase "This work
presents a very low-cost solution for irrigation automation based on the cloud..”

The sentence has been modified: “This paper presents a low-cost solution for automatic cloud-based irrigation”

·         Comment 2: “Define the color usage in Fig. 2”

A colour legend has been added to explain the figure 2 (line 118). “Blue”= Live Coverage; “Purple”= Under roll-out.

 ·        Comment 3: “Explain in more detail the usage of both LoRA and SIGFOX (Local star topology for LoRA and Sigfox for the LoRA GATEWAY?) The solution presented here uses LoRa communications at the local level, and SIGFOX communications to access IoT platforms..”.

A deep explanation has been added in Section 2.1.1: “Thus, the result of communication consists of a LoRa network for the local level. Figure 7 shows the local nodes (hydrants and monitors) connected by LoRa technology (star topology) to a gateway, consisting of a LoRa module. The connection to the cloud is detailed in section 2.1.5 and 2.2.”  (lines 211-213) and in Section 2.1.5 this sentence has been added: “The module MKR1200 communicates to the cloud through SIGFOX”.

·         Comment 4: correct the phrase "of a comprehensive monitoring and performance system..."you mean "high performance"?

The sentence has been modified: “implementation of a comprehensive monitoring and a high performance system based on communication”.

·         Comment 5: “maybe the phrase "The IoT platform used in this work (ThingSpeak) integrates the MATLAB® tool which allows
the implementation of analysis and decision-making programs.."
must be removed from the conclusions section to other section of the article.”

We have removed this sentence from Conclusions (an empirical conclusion has been inserted: “The IoT platform used in this work (ThingSpeak) has allowed interaction with controlled facilities from any device connected to the Internet”) and the original one has been moved to section 2.2.
